# Parsing the Old Using the New and Vice Versa: Fine-tuning Pre-Trained Language Models for Icelandic

Affiliation / Address line 1
Affiliation / Address line 2
Affiliation / Address line 3
`email@domain`

Affiliation / Address line 1
Affiliation / Address line 2
Affiliation / Address line 3
`email@domain`

## Abstract

In this study, we present experiments on parsing historical Icelandic by using a pre-trained ConvBERT language model for modern Icelandic which is then fine-tuned on modern Icelandic, historical Icelandic, and on a combination of both. Using the dependency parser DiaParser, the models are evaluated on both modern and historical Icelandic. The results indicate that fine-tuning on in-domain data is ideal; fine-tuning on historical texts when parsing historical texts achieves 82.9% LAS, and fine-tuning and testing on modern text reaches 87.96% LAS. The best performing model is obtained on fine-tuning on a merged dataset, achieving 85% LAS on historical data, and 89% on modern data.

## 1 Introduction

We have seen an impressive development of language resources and tools for written and spoken Icelandic during the past 15 years, including corpora of historical and modern Icelandic and pipelines of tools for automatic morpho-syntactic annotation, among others. The tools developed over the years are built upon the current state-of-the-art methods of the time they have been produced, from finite-state techniques to statistical and neural modeling. Recently, the emergence of large pre-trained general language models have been also applied to Icelandic as a consequence attributed to the rise in popularity of transformer models and their state-of-the-art performances on many downstream NLP tasks. In the past year, such a neural pipeline for parsing Icelandic has emerged (Arnardóttir and Ingason, 2020). The pipeline has been tested on the Icelandic Parsed Historical Corpus (IcePaHC) (Rögnvaldsson et al., 2012), a diachronic treebank consisting of modern and historical text of around one million words. The reported F1 score was 84.74%, precision 85.07% and recall 84.43% when trained on the multilingual BERT. The authors reported a pilot study using old vs modern data in the training and test sets and concluded that varying the data did not increase the parsing performance.

The purpose of this study is to explore the impact of the data origin in terms of its age on the parsing quality of historical and modern Icelandic. We carry out experiments on historical Icelandic texts and modern Icelandic texts using a language model pre-trained on modern Icelandic, that is then fine-tuned on modern versus historical texts, respectively. We then evaluate the parsing quality on the different combinations of training data (fine-tuning) and test the models on historical vs modern Icelandic texts. We seek to find out if it is necessary to fine-tune a model on historical texts to achieve an adequate quality parsing of historical texts, or might it as well be fine-tuned on modern Icelandic to expend less high value historical data. We also want to see if the same holds for modern texts. In addition, we merge the historical and modern data to see how the combined dataset can improve parsing performance.

The experiments as part of this research are inspired by the work done by Guðnason and Loftsson (2022) and utilise the same pre-trained state-of-the-art language model (ConvBERT(Jiang et al., 2020a)) and parser software (DiaParser(Attardi et al., 2021)) as was used in their research.

The structure of the paper is as follows. In Section 2, an overview of the field is introduced, first describing Icelandic with its language resources and tools of relevance to our study, then describing the parsing framework applied in our study. In Section 3, we present the data used in our experiments, and in Section 4, we describe the method and experimental setup. We present the results in Section 5, and discuss them in the subsequent sec-

tion. Finally, we conclude our main findings and suggest some future work in Section 7.

## 2 Related work

### 2.1 Historical and Modern Icelandic

Parsing historical texts involves many challenges due to the diachronic change of morphology and syntax which can spell trouble for parsers developed for modern languages.

Many languages, including the continental Nordic languages, have seen major changes in grammar throughout the last couple of centuries. The same cannot really be said for Icelandic which is a language that, much thanks to the uniquely isolated geography of the country and its strong literary tradition, has retained much of its morpho-syntactic features from Old Icelandic (Rögnvaldsson et al., 2012). Even though much of the language's grammar has been retained for such a long time, it has, however, seen a few changes throughout the years (Rögnvaldsson and Helgadóttir, 2011; Rögnvaldsson, 2005)

Some of the major linguistic changes include differences in word order, mainly in the verb phrase, the introduction of modal constructions (vera að, vera búinn að), and an expletive equivalent of the word "it"/"there"(það). However, even though Icelandic, similar to other languages, evolved over time, the grammar of the Icelandic language remained relatively stable.

### 2.2 Icelandic LT

Despite the fact that the Icelandic language only has approximately 350.000 native speakers, both language tools and resources for Icelandic has seen extensive development, especially during the last decade. In 2017 the Icelandic government decided to fund a language technology program in order to increase the availability of the language in language technology related software and applications (Nikulásdóttir et al., 2020).

However, it was already back in the year of 2012 when a treebank consisting of syntactically parsed Icelandic historical texts was published, called IcePAcH (Rögnvaldsson et al., 2012). The corpus contains approximately one million words from texts covering a span of almost an entire millennium, ranging all the way from the 12th century up until the 21th century. The annotation scheme for the original treebank is closely following the scheme used in The Penn Treebank (Rögn-valdsson et al., 2012) but has since also been converted to the Universal Dependencies(UD) annotation scheme (de Marneffe et al., 2021). The intended use of this diachronic corpus was, first and foremost, to act as a language technology resource and as a tool for syntactic research of the Icelandic language. It has since been one of the go-to data resources for many NLP-projects, including the morpho-syntactic analysis of Icelandic.

### 2.3 Parsing Icelandic

Several efforts have been made over the years to build systems for syntactic analysis of Icelandic. The first published parser for Icelandic is the so-called IceParser, an incremental, shallow finite-state transducer parser developed by Hrafn Loftsson (Loftsson and Rögnvaldsson (2007)).

A more recent parser developed by Þorsteinsson et al. (2019) presented a wide, context-free grammar and an Earley-based parser (referred to as Greynir). In contrast to the IceParser, this parser fully parses sentences and does so on the basis of the accompanying handcrafted context-free grammar.

Last but not least, there is also a pipeline for parsing plain Icelandic text using the IcePaCH annotation scheme presented by Arnardóttir and Ingason (2020). The model is a development of the non-neural pipeline by Jökulsdóttir et al. (2019). The former one uses the Berkeley Neural Parser and achieves high accuracy at a high speed using a multilingual BERT model. Interestingly, Arnardóttir and Ingason (2020) also did some minor experiments by splitting the train, development and test data of the IcePaCH treebank by age so that the oldest 80% was used for training and the rest for test, and vice versa. These results were, according to themselves, "unfavorable" as the foremost achieved an F1 score of 77.01 and the latter a score of 82.57, in comparison to their main experiment achieving an F1 score of 84.74 by training and evaluating on an even mix of the IcePaCH texts.

As transformer-based language models became not only popular, but also the norm for many languages thanks to their state-of-the-art performances in various down-stream NLP tasks, it was only inevitable before they would come to be evaluated on Icelandic as well. In their paper *Pre-training and Evaluating Transformer-based Language Models for Icelandic*, Guðnason and Lofts-

son (2022) describe their research on such models for various down-stream tasks, among them parsing. During their research, they pre-trained four different transformers-based language models on a large quantity of Icelandic data from the Icelandic Gigaword Corpus[1] containing 1.69 billion tokens from a vast variety of different domains. More than 94% of the texts in this corpus date back to the period after the year 1980, while some of the text domains contain texts that date back to the 13th century (Steingrímsson et al., 2018).

These large general purpose language models were later fine-tuned to perform 4 different down-stream tasks. For the task of parsing, Guðnason and Loftsson (2022) fine-tuned each one of these models on the aforementioned IcePaCH-treebank converted to the Universal Dependencies[2] annotation scheme. The training and evaluation was done using the open-source dependency parser called DiaParser based on bidirectional long-short term memory, BiLSTM-based biaffine, utilising transformers. In this case the transformers purpose is to extract contextualised word representations (Attardi et al., 2021). When Guðnason and Loftsson (2022) later evaluated using Label Attachment Score (LAS) for the parsing part of their research, they came to the conclusion that, out of the languages models they tested, the so-called ConvBERT-Base transformer-based model, with a LAS score of 86.50%, performed the best by a slight margin. In the end they conclude that the transformer-based language model outperforms previous state-of-the-art models on similar tasks for Icelandic. This pre-trained ConvBERT-base model was given the name convbert-base-igc-is[3]. Next, we describe the experimental setup of our study starting with the description of the dataset used.

## 3 Data

The data used in the experiments of this study comes from a mix of three treebanks, all annotated according to the Universal Dependencies scheme (Kübler et al., 2009) and (de Marneffe et al., 2021). The first, and by far the largest treebank used is the aforementioned UD converted IcePaCH treebank[4] which contains around 1 mil-

lion tokens of text, ranging all the way from the 12th century up until the 21st century. There is a fairly even distribution in text representation from each of the covered centuries, although the amount of text from the 21st century is very limited. The treebank covers a surprising amount of different domains ranging from sagas/fiction to science, religious texts, biographies and legal texts.

In addition, a large chunk of modern texts were later added to the IcePaCH treebank and released as the UD Icelandic Modern[5] which we also included in our study. The treebank contains an additional 80.000 tokens of Icelandic texts from the 21st century. These texts come from unprepared parliament speeches and sports news reports.

As a last addition to the modern side of the data, the UD Icelandic PUD treebank[6] is also used which contains an additional 1000 sentences of morphologically and syntactically annotated Icelandic translations of texts from news and from Wikipedia.

Table 1 lists the total number of sentences and tokens for each dataset.

| Treebank | Sentences | Tokens |
|----------|-----------|--------|
| IcePaCH | 44 029 | 873 743 |
| UD Modern | 6 928 | 145 322 |
| UD PUD | 1 000 | 16 869 |

Table 1: Treebank sizes in number of sentences and tokens

In order to run the intended fine-tuning experiments, the data from all three treebanks was divided into a historical subcorpus and a modern one. The most convenient way of doing this would be to simply treat the IcePaCH treebank as the historical part and the other two treebanks as the modern part, but there are two problems to this solution. Since the IcePaCH treebank contains texts ranging all the way up until the 21th century, not all of these texts can be considered historical. Also by doing so the historical part would be much larger than the modern part. Hence the data was divided by defining a point in time from where the texts can be considered modern. This point in time was decided to be the beginning of the 20th century, where texts older than this are considered his-

---

[1] https://clarin.is/en/resources/gigaword/

[2] https://universaldependencies.org/

[3] https://huggingface.co/jonfd/convbert-base-igc-is

[4] https://github.com/UniversalDependencies/UD_Icelandic-IcePaHC, last retrieved on 26th October 2022

[5] https://github.com/UniversalDependencies/UD_Icelandic-Modern, last retrieved on 26th October 2022

[6] https://github.com/UniversalDependencies/UD_Icelandic-PUD, last retrieved on 26th October 2022

torical and texts written later are considered modern. Where to draw this diachronic distinction is a challenging task, but for the purpose of this study this simple point in time was deemed appropriate as to separate recent use of the language with older use of the language.

Since the historical subcorpus was still much larger than the modern one, the historical part was stripped off some sentences as to mach the size of the modern part. This was done so that the experiments would be more comparable as it would otherwise be hard to draw conclusions if the experiments are not using the same amount of fine tuning data. Table 2 presents the size of the revised datasets in tokens, approximately following the traditional 80-10-10 split for the historical and modern data, followed by the number of sentences used in each.

| Token | train | dev | test | total |
|---|---|---|---|---|
| Hist | 224 782 | 29 975 | 38 492 | 293 249 |
| Mod | 232 809 | 40 553 | 36 104 | 309 466 |
| Sent | 12 941 | 1 579 | 1 804 | 16 324 |

Table 2: Data split in number of tokens and sentences (Sent) for historical (Hist) and modern (Mod) texts.

The three treebanks did not have to be cleaned or pre-processed; the UD format fitted perfectly to the tools used in our experiments, described next.

## 4 Method

To parse the Icelandic texts, we use a pre-trained language model, which is fine-tuned on the modern and the historical datasets. To carry out the experiments and allow comparison to the study which this paper has been heavily inspired by (Guðnason and Loftsson, 2022), we chose the same set of tools and pre-trained language model for easier comparison. First, we describe the choice of the pre-trained language model, then the parser used for fine-tuning and evaluation. Then, we explain the fine-tuning process, the evaluation, and the experimental setup in detail.

### 4.1 Language model

There are many variants of the basic pre-trained BERT-model out there, many of which have performed at a state-of-the-art standard for many NLP tasks. The basic concept behind BERT is pre-training on unlabeled data through different tasks

and then fine-tuning the model to perform more specific tasks (Devlin et al., 2019). Yet another variant, ConvBERT, was proposed by Jiang et al. (2020b) as an improvement to the basic BERT-model to improve on some of its limitations as the basic model relies on a global self-attention block at the cost of computational efficiency and a large memory footprint. In ConvBERT, this self attention mechanism has been enhanced with a span-based convolutional module that significantly helps with the computational efficiency of the model and also increases performance on some downstream NLP tasks (Chang et al., 2021; Guðnason and Loftsson, 2022). We use the aforementioned convbert-base-igc-is which has been shown to outperform all pre-trained models on the downstream task of parsing when trained on the unlabelled Icelandic Gigaword Corpus Guðnason and Loftsson (2022).

### 4.2 DiaParser

Both the fine-tuning of the convbert-base-igc-is model and later on the evaluation was done using the DiaParser, which is, as of 2021, a state-of-the-art transformer-based dependency parser that builds upon the architecture of the Biaffine Parser by Dozat and Manning (2016). The parser derives wordpieces from an input using a transformers based tokenizer and then obtains word embeddings by averaging over the wordpiece embeddings (Attardi et al., 2021).

The parser utilises the HuggingFace Transformers API and allows the user to specify which available language model they want to use from the HuggingFace website. In the case of this study, this is used to access the convbert-base-igc-is model. The code for the parser is freely available on GitHub[7].

Although the DiaParser is a relatively recent piece of parser software (last updated in April 2021) there are some parts of the code that are not compatible with the most recent updates of the packages required by the software. To fix this, the variable name 'max_length' in the code for the parser was replaced with the name 'model_max_length' in order to conform to an update made in the AutoTokenizer library for Transformers. An older version (4.19.2) of the 'Transformers' library also had to be installed in order to make the parser operable.

---

[7]https://github.com/Unipisa/diaparser

### 4.3 Fine-tuning

During the fine-tuning of the model, the provided configuration file[8] containing all training parameters was kept unchanged with the exception of changing the default transformer model to the convbert-base-igc-is model. In other words, this means that the training was done using the default settings and was done until convergence for the training sessions. When the model is converging, the DiaParser automatically detects this, aborts the training process and saves the model checkpoint that achieved the highest dev-score.

The models converged early on throughout the training process where the model fine-tuned on the historical data converged around epoch 100 and the model fine-tuned on the modern data converged somewhat later at around epoch 160 as indicated in Figure 1. This is out of a maximum of 1000 epochs.

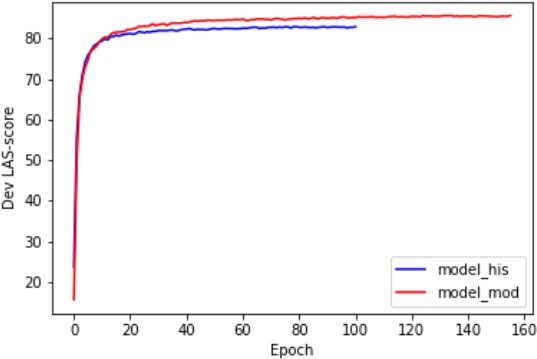

Figure 1: Convergence during training

The training was done using a single Nvidia T4 GPU via the Uppmax cluster where one training session took approximately 2 hours to complete.

### 4.4 Evaluation

When the training was completed, each, now fine-tuned, model was evaluated on the test set for both sides of the data using DiaParser's evaluation function. The evaluation was performed on a local CPU since it does not require as much computing power. The standard Labeled Attachment Score (LAS), i.e. the percentage of words that get the correct head and label was used for metrics.

### 4.5 Experimental setup

To test how fine-tuning on historical and modern data performs on both types of texts, we first fine-

---

[8]https://github.com/Unipisa/diaparser/blob/master/config.ini

tuned the pre-trained convbert-base-igc-is model on the historical vs the modern data, and then evaluated the parsing quality on the modern and the historical test sets using Labeled Attachment Score (LAS). Figure 2 illustrates the process.

In total there were two sessions of fine-tuning, one on each type of the dataset, and four sessions of evaluation, twice on each of the two test sets using the two different fine-tuned models derived from the first step. In other words, the model that was fine-tuned on the historical subset of the data was later evaluated on both the historical and the modern part of the data. The same evaluation procedure was then replicated using the model fine-tuned on the modern set of the data.

In addition, we also carried out fine-tuning on the merged historical and modern datasets to see how the increase of the dataset size would affect parsing performance.

## 5 Results

To serve as a baseline for the experiments, the best result from Guðnason and Loftsson (2022) is used. The motivation for this is that the dependency parsing experiments conducted during their research is what all experiments conducted in this study are inspired by. They use the same pre-trained model and the same parser software to achieve their best LAS which is 86.50%. Since they both fine-tuned as well as evaluated on the same dataset (IcePaCH), which according to the definition of historical and modern data made as part of this study contains both historical and modern texts, this baseline seems extra relevant and will serve as a great point of reference.

In Table 3 the LAS from the baseline and all experiments are listed. The results show that the best result achieved by the non-merged models (87.96 LAS) is obtained on the modern dataset using the model fine-tuned on the same, modern type of text, which also outperformed the baseline (86.5 LAS). Not surprisingly, the results also show that the model fine-tuned on the historical data scored the highest on the historical test data and scored significantly worse on the modern test data with 77% LAS. Thus, it is clear that the type of data used for fine-tuning is important when parsing historical and modern Icelandic, since the difference is more than 10 percentage points between the best and the worst non-merged model. It is, however, worth noting the reasonably high LAS obtained by

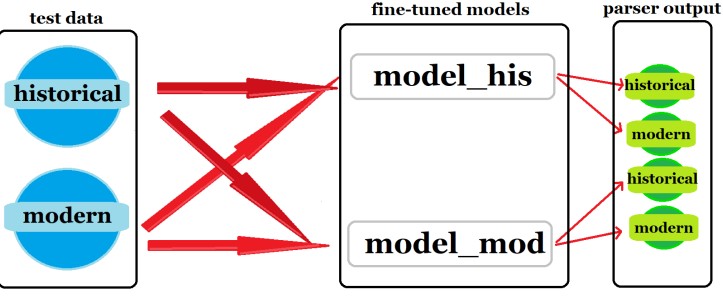

Figure 2: Visualised method. Test data is passed through the two models to yield 4 outputs.

the modern model on the historical dataset as the score is only 3 percentage points lower then the LAS obtained by the historical model on that same test data.

Lastly, and least surprisingly, the best overall result on both test sets was achieved by the merged model, achieving 85.04 LAS for the historical test and 89.07 LAS for the modern side.

# 6 Discussion

By simply looking at the LAS from the experiments it is possible to spot the most obvious, and not too surprising pattern, namely that a model, fine-tuned on one type of the data, performs better when evaluated on that same type of the data. In other words, the in-domain model performs better on in-domain data. So what are the mistakes that the out-domain model does that the in-domain model does not?

First of all, a linguistic, qualitative manual evaluation of both in-domain datasets revealed that neither of the two models performed flawlessly. For longer sentences, the in-domain model usually struggles to connect the correct syntactic head to the right word which then naturally leads to a mislabeling of the syntactic relation. For shorter sentences of in-domain data, the in-domain model performs much better and usually manages to successfully connect the correct syntactic heads to the correct words. In those cases when it does make mistakes for shorter sentences, it is usually in the form of mislabeling relations. Both of these mistakes are even more prevalent for out-domain models, hence the worse results. However, for many shorter sentences where the in-domain model was very close to correctly parse the entire sentence, the out-domain model sometimes struggled with the correct labels. This was the case even if all of the syntactic heads were correctly assigned as displayed in Figure 3.

Another discrepancy found in the results is the difference in performance between the models on their respective in-domain data. The model fine-tuned on the modern data performed much better on its in-domain data than what the model fine-tuned on the historical data did on its own in-domain data. One explanation for this might simply be due to the fact that the pre-trained convbert-base-igc-is model was pre-trained on mostly modern Icelandic texts and hence should be a better fit for tasks concerning modern data. This could also explain why the modern model evaluated on the modern data even outperformed the baseline with a margin of more than 1.5 percentage points since the baseline was achieved through fine-tuning and evaluation on the IcePaCH corpus, or in other words, on a majority of historical texts. The baseline is more comparable to the results given by the historical model evaluated on the historical data since this result was achieved in a very similar manner to how the baseline was achieved. However, as previously mentioned, the IcePaCH corpus contains not only historical texts but also some modern additions. Hence, fine-tuning and evaluating the convbert-base-igc-is model on this corpus without any further data splits (as was the case for the baseline) should be beneficial to this model in comparison to, for instance, the historical model evaluated on the historical data. This is the case since this model had nothing to do with any modern data other than from the pre-trained language model itself. This could be an explanation of why the baseline outperformed the historical model.

Another important point to bring up when discussing potential factors as to why the historical model performed worse overall is the fact that the Icelandic texts that this model was fine-tuned and evaluated on are taken from a much larger diachronic span than the texts for the modern data. The texts span from the 12th century up until the

| Model | IcePaCH | | test_historic | test_modern |
|---|---|---|---|---|
| Model | IcePaCH | model_his | 82.92% | 77.04% |
| baseline | 86.50% | model_mod | 80.34% | 87.96% |
| | | model_hismod | **85.04%** | **89.07%** |

Table 3: LAS from the baseline and evaluation of the models

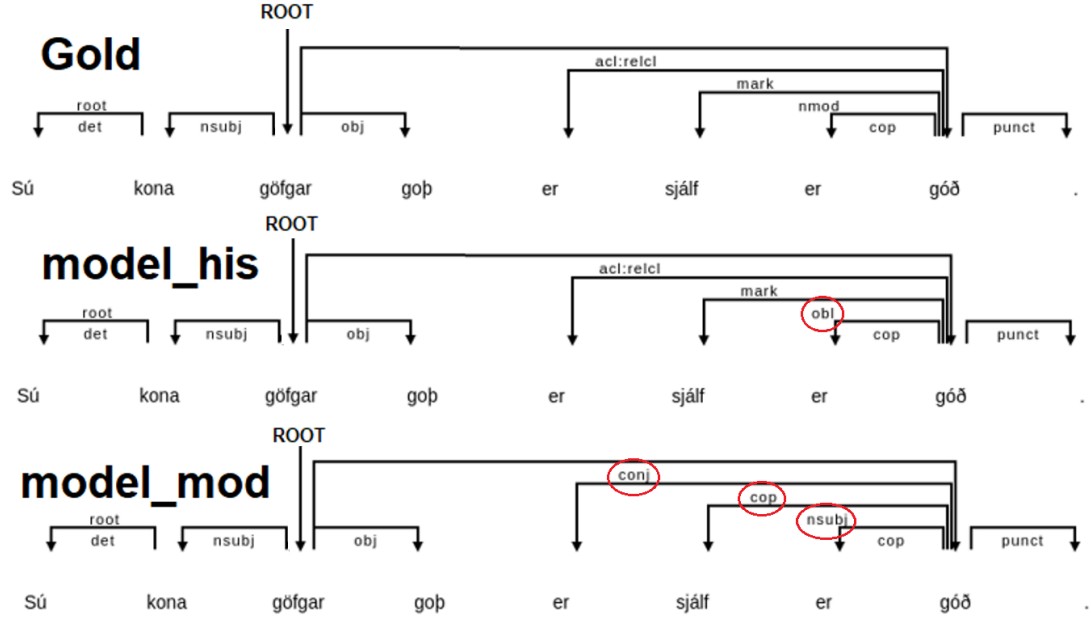

Figure 3: Comparison between parsing predictions on a historical sentence. Mislabeled relations are encircled in red

20th century which is a long time under which the language would have the time to change. Hence, there is bound to be more variation in syntax between the texts covered in the historical side of the data as compared to the modern side in which texts only span from a time of a little more than a hundred years.

It is also worth mentioning that, even though the historical and the modern domain were the focal point of this research, there is also plenty of variation within each one of these two domains. Like mentioned in Section 3, the data comes from various sources/domains like sagas, religious texts, legal texts, parliament speeches etc. and this could of course have an effect on the final results.

## 7 Conclusion

This research set out to explore parsing quality on historical and modern Icelandic texts using pre-trained language models that are fine-tuned on either historical or modern Icelandic texts. The intent was to answer the question of whether or not

it is necessary to fine-tune on in-domain texts to parse in-domain texts, or if it is feasible to replace the in-domain texts with out-domain texts for training. In this study, the domains considered were historical and modern Icelandic. In addition, we also run experiments on a merged model consisting of both the historical and modern dataset.

In the end, the results suggest that, when parsing historical Icelandic, ideally, fine-tuning a pre-trained Icelandic language model on historical texts is the preferred option. This suggestion also holds the other way around. However, given the relatively high performance by the modern model when evaluated on the historical data, one could also make the suggestion that this could also be a viable option. The same, however, cannot be said for using historical data to fine-tune a model to parse modern Icelandic as the results suggest otherwise. It is important to keep in mind that this could be the result of modern Icelandic bias given the pre-trained language model. There is, although, much more work that needs to be put into this topic to draw a more general conclusion.

Future research could investigate time-specific data by centuries or other time periods of interest to see whether LAS could be increased by fine-tuning on in-domain, time specific datasets. Also, other deep learning approaches could be tested to see if LAS could be further increased. Last but not least, an in-depth evaluation of the labeled attachments scores, the relations between the heads and their dependents would be beneficial.

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
