# OpenReview forum: "Parsing the Old Using the New and Vice Versa: Fine-tuning Pre-Trained Language Models for Icelandic"
_NoDaLiDa/2023/Conference — NoDaLiDa 2023_

### Official Review · Reviewer_Z3Gd · 2023-02-19
**A student paper that is not very original and has quality issues, but is significant for the Nordic LT community and overall the text is clear.**

**Rating:** 6
**Confidence:** 4

**Review:**

This paper describes experiments with fine-tuning a large Icelandic language model (LM) on dependency parsing data with the purpose of evaluating the performance on both historical and modern Icelandic.  The fine-tuning and evaluation is fourfold: 1) fine-tuning and evaluation on modern data; 2) fine-tuning on modern data and evaluation on historical data; 3) fine-tuning and evaluation on historical data; and 4) fine-tuning on historical data and evaluation on modern data.

The answer to the research question put forth, i.e. whether it is necessary to fine-tune a model (pre-trained on modern data) on historical texts to achieve an adequate quality parsing of historical texts, is kind of obvious, but, nevertheless the work is interesting because it also presents evaluation results for the case of merged (modern and historical) data.

The first author is a student and the text of the paper reflects that in various places (for example, the lack of support of claims). The main problem, however, is that it seems that the paper has not been proofread adequately (a quality issue).  The work is not very original, but it is significant for the Nordic LT community and overall the text is clear.

I feel that the paper should be accepted given that the text is fixed according to the detailed list of comments below.

Citations and bibliography
--------------------------
All entries in the bibliography need to be checked for correct title casing. One example is:
A neural parsing pipeline for icelandic using the berkeley neural parser -> A Neural Parsing Pipeline for Icelandic Using the Berkeley Neural Parser

"We have seen an impressive development of language resources and tools ..." -> you need to provide references to support this claim.

"Recently, the emergence of large pre-trained general language models have been also applied to Icelandic ..." -> you need to provide references to support this claim.

In several places in the paper, you refer to "Guðnason and Loftsson (2022)". The is incorrect and should be "Daðason and Loftsson", see https://aclanthology.org/2022.lrec-1.804/

"The annotation scheme for the original treebank is closely following the scheme used in The Penn Treebank (Rögnvaldsson et al., 2012)" -> Here the reader is mislead to believe that Rögnvaldsson is the author of the Penn Treebank!

The following entry needs to be fixed (missing spaces) and an English translation of the title of the article would be appropriate to add in parenthesis:
Setningafræðilegar breytingar í íslensku. Setningar. Handbókumsetningafræði. Íslensktunga3, pages 602–635.

"It has since been one of the go-to data resources for many NLP-projects, including the morpho-syntactic analysis of Icelandic." -> you need to provide references to support this claim.

"The first published parser for Icelandic is the so-called IceParser, an incremental, shallow finite-state transducer parser developed by Hrafn Loftsson (Loftsson and Rögnvaldsson (2007))."  -> "The first published parser for Icelandic is the so-called IceParser, a shallow parser based on incremental finite-state transducers, developed by Loftsson and Rögnvaldsson (2007)."

There are missing first names in the following entry:
TF Jökulsdóttir, AK Ingason, and EF Sigurðsson. 2019. 814 A parsing pipeline for icelandic based on the icepahc 815 corpus. In Proceedings of CLARIN Annual Confer- 816 ence, pages 138–141.

"All annotated according to the Universal Dependencies scheme (Kübler et al., 2009) and (de Marneffe et al., 2021)." -> "All annotated according to the Universal Dependencies scheme (Kübler et al., 2009; de Marneffe et al., 2021)."

"which has been shown to outperform all pre-trained models on the downstream task of parsing when trained on the unlabelled Icelandic Gigaword Corpus Guðnason and Loftsson (2022)." -> "which has been shown (by Daðason and Loftsson (2022)) to outperform all pre-trained models on the downstream task of parsing when trained on the unlabelled Icelandic Gigaword."


Clarifications/fixing needed
----------------------------
"The reported F1 score was 84.74%, precision 85.07% and recall 84.43% when trained on the multilingual BERT."  Do you mean that mBERT was fine-tuned using IcePaHC?  You should also provide a reference to mBERT.

"... or might it as well be fine-tuned on modern Icelandic to expend less high value historical data."  Not clear what is meant by "to expend less high value historical data"

"The model is a development of the non-neural pipeline by Jökulsdóttir et al. (2019)." -> What is meant by "development of"?

"also did some minor experiments" -> do you mean "preliminary" when you write "minor"?

"the historical part was stripped off some sentences as to match the size of the modern part."  How were these sentences selected?

"Yet another variant, ConvBERT, was proposed by ..."  No other specific variants have earlier by mentioned!

"The training was done using a single Nvidia T4 GPU via the Uppmax cluster ..."  Should provide a footnote about Uppmax

You introduce the abbreviation "Labeled Attachment Score (LAS)" in several places in the paper.

Table 3 is too far away from the Results section.  I suggest moving the table immediately after Figure 2.


Spelling, grammar and minor issues
----------------------------------

Abstract:
"The best performing model is obtained on fine-tuning on a merged dataset" -> "The best performing model is obtained when fine-tuning on a merged dataset"

Intro:
"The tools developed over the years are built upon .." -> "The tools developed over the years were built upon .."
" ... have been also applied to Icelandic ..." -> " ... have also been applied to Icelandic ..."
"... of around on million words." -> "... of about on million words."
"... using old vs modern data" -> "... using old vs. modern data"
"... that is then fine-tuned ..." -> "... which is then fine-tuned ..."
"... on historical vs modern Icelandic texts" -> "... on historical vs. modern Icelandic texts"
"... by the work done by" -> "... by the work carried ot by"
Add missing spaces to references in the paragraph that starts with "The experiments as part of this research ..."

Related work:
(vera að, vera búinn að) -> ('vera að', 'vera búinn að')
In general, you should use LaTex quotes, i.e ``for opening double quotes and '' for closing quotes, and similar for single quotes.
"has approximately 350.000 native speakers" -> "has approximately 350,000 native speakers"
"both language tools and resources for Icelandic has seen extensive development," -> "have seen"
"However, it was already back in the year of 2012 when a treebank ..." -> "However, already in the year of 2012 a treebank ..."
"ranging all the way from the 12th ..." -> "ranging from the 12th ..."
"the Universal Dependencies(UD)" -> "the Universal Dependencies (UD)"
"A more recent parser developed by Þorsteinsson et al. (2019) presented a wide ..." -> "A more recent parser, developed by Þorsteinsson et al. (2019), presented a wide ..."
"the oldest 80% was used for training and the rest for test ..." -> "for testing"
"as the foremost achieved an F1 score of 77.01 and the latter a score of 82.57" -> "the former" (not "the foremost")
"down-stream tasks" -> "downstream tasks"
"In this case the transformers purpose is to ..." -> ""In this case, the transformers purpose is to ..."

Data:
"ranging all the way from the 12th century up until the 21st century" -> This was already mentioned in Section 2.2
"The treebank contains an additional 80.000 tokens of Icelandic ..." -> "The treebank contains an additional 80,000 tokens of Icelandic ..."
The additional info in footnotes "last retrieved on 26th October 2022" is not necessary
"but there are two problems to this solution" -> "but there are two problems to this approach"
"if the experiments are not using the same amount ..." ->  "if the experiments were not using the same amount ..." ->

Method:
"To carry out the experiments and allow comparison to the study which this paper has been heavily inspired by (Guðnason and Loftsson, 2022), we chose the same set of tools and pre-trained language model for easier comparison." -> "To carry out the experiments and allow comparison to previously published results, we chose the same set of tools and pre-trained language model as Daðason and Loftsson (2022)."
"Then, we explain the fine-tuning process, the evaluation, and the experimental setup in detail." -> "Finally" instead of "Then"

Conclusion:
"In addition, we also run experiments ..." -> "ran experiments ..."
"In the end, the results suggest that ..." -> "Our results suggest that ..."




**Paper Type:**

Long paper

---

### Official Review · Reviewer_xffa · 2023-03-07
**Fine-tuning with old vs modern texts for syntactic parsing**

**Rating:** 6
**Confidence:** 4

**Review:**

In this paper, the authors investigate fine-tuning settings for parsing modern and old texts in Icelandic. They propose to explore the impact of using modern vs old data on parsing quality, to check whether the use of old data is necessary to get good performance on this type of texts. They also explore the reverse scenario (but the PLM is always pretrained on modern texts).

The authors use three corpora: the largest one contains a mix of modern and historical texts, covering almost an entire millennium and varied genres, and it is merged with small corpora of modern texts, all annotated within the UD framework. They split the data into modern and historical choosing the beginning of the 20th century as splitting point.

The experimental setting is based on the use of a PLM pretrained on modern Icelandic (6 % of the data are old texts, however), fine-tuned on either modern or old texts, and then evaluated on the parsing task on both modern and old texts. A final experiment use a fine-tuning on a merge corpus of old and modern texts. The experiments show that fine-tuning within the same ‘age’ makes for better results, as expected (10 % of difference in average), with however rather good performance for the model fine-tuned on modern data and test on historical ones, with only 3 % decrease compared to the one fine-tuned on historical data. Finally, the merged model allows to get the best performance on both datasets. A manual analysis reveals that out-of-domains models often struggle with labelling the right relations, even when the attachments are right. The authors note that their models do not perform very well on historical texts, which could be due to the fact that the PLM is pretrained on mostly modern texts.

This paper presents experiments on parsing Icelandic texts, in a diachronic setting. The goal is interesting: demonstrating the robustness of syntactic parsers across language evolution. The setting is sound and the paper mostly clear. I think this paper could be interesting for the NODALIDA community. I have however a few remarks, and I think the paper could be made stronger with a few more experiments. The paper has also some clarity issues, detailed below.

- I’m not sure to understand why the authors chose the beginning of the 20th century to split between historical and modern. To me, what we consider old texts are older than that: are the syntactic differences presented in the paper between old and modern Icelandic relevant for this distinction? As said in the end of the paper, it’s a long period for the historical texts, with possibly a lot of variations. Maybe it could be interesting to test other splits, maybe into 3 parts rather than 2?

- In order to have more balanced data, the authors removed some historical texts: how much of the historical data were removed? I think that it would be interesting to have experiments demonstrating that augmenting the number of data indeed improves the performance (or not), and thus also to have performance using all the available data. In the same way, trying to fine-tune on a merging of historical and modern data while keeping the same amount of data as for the separate models would have been interesting to show possible improvement in a form of a transfer.

- The influence of the pretraining on modern data is unclear: would it be possible to retrain the model? If not please say it explicitely in the paper.

- The comparison with the previous system in unclear: the test set is not the same right? could you please be more explicit about this comparison?

Clarity / typos:
- The intro does not make it clear the contributions of this paper compared to (Rögnvaldsson et 2012) who also test varied amount of old vs modern texts, and compared to (Guönason and Loftsson, 2022) who are said to use the same PLM and parser.
- The claim isn't clear, why fine-tuning on old texts could be useful / helpful for modern texts?
- please be more precise in the caption of the table, i.e. ‘model_his’ = model fine-tuned on historical data ...
- l. 312 : 21th → 21st


**Paper Type:**

Long paper

---

### Official Review · Reviewer_Q4MP · 2023-03-09
**Good evaluation paper on parsing modern Icelandic training on historical material and vice versa.**

**Rating:** 8
**Confidence:** 4

**Review:**

The authors look at training a dependency parser for Icelandic by finetuning a BERT-like model. In particular, they look at domain effects, where "domain" here is defined by temporal difference: training on historical material and applying to modern material and the other way around. The paper is very clear, well-written and has a clearly defined goal.

The authors show that (unsurprisingly) training and testing on material from the same "period" works best, but also that training on modern material and applying to historic material works fairly well.  The reverse scenario -- training on historical and applying to modern -- fares the worst. Investigating the feasibility of using modern parsers on historical material is important, since it is typically in this direction that there is resource-scarceness: there is more development material available for the modern stage. Interestingly, the authors also show that lumping the data together works well for both periods, which is likely to be due to the advantage of having more training data, even if it is out-of-domain.

There are some points where the paper could be improved:
- The comparison to the base line given is relevant, but since -- if I understand it correctly -- we are not talking about **exactly** the same evaluation sentences, perhaps not to big a deal should be made about surpassing the baseline or not. But for use as a ball-park figure of what is the current state of parsing Icelandic it is valuable.

- I would like to know more about how the historical material was pruned to make a dataset that is as big as the modern dataset. Also, it is as big in terms of sentences, but not in terms of words/tokens. Especially the dev set is smaller for the historical material tokenwise where does the drastic difference come from? Does it impact parser performance?

- A scenario that would be open to people developing a parser for a resource scarce domain, like historical material in many cases, is to have training material from the rich domain (modern material) but a dev set from the resource-poor domain (historical). This is a case that I miss here: what happens if you train on modern material with a historical dev set and evaluate on the historical material? (Interestingly that the general strategy of having dev material that is like the test material is presented as good praxis here https://cs230.stanford.edu/blog/split/)

- The use of figures in the paper could improve: make sure to discuss the figures that you show. Eg the parse trees shown take up half a page (not very effective use of page real estate!) but are discussed in only a few sentences. More could be made of this. Perhaps discuss some stats over the kind of mistakes instead? (In general, the error analysis I think was the relatively weakest part of an otherwise very clear and good paper.)

- It would be good to mention some properties of the IcePaHC, for one the spelling of the texts have been modernized, so spellingwise a "rendition" of the historical texts in modern Icelandic spelling. Also, the sentence segmentation has been performed. This means that two hard problems when dealing with historical text -- 1) form variation because of spelling and 2) the identification of units to parse, which typically cannot rely on punctuation as it can in modern Icelandic (English / French / etc) text -- have already been solved before the material gets to the parser.


**Paper Type:**

Long paper

---

### Decision · Program_Chairs · 2023-03-17

**Decision:**

Reject

**Comment:**

The authors informed the PC:
I wish to inform you that we have decided to withdraw our submission "Parsing the Old Using the New and Vice Versa: Fine-tuning Pre-Trained Language Models for Icelandic" from the NoDaLiDa conference as it has come to our attention that there were duplicate sentences in the version of the corpus that we used. This issue has since been solved by the creators of the corpus.